# Net Optical Gain Coefficients of Cu^+^ and Tm^3+^ Single-Doped and Co-Doped Germanate Glasses

**DOI:** 10.3390/ma15062134

**Published:** 2022-03-14

**Authors:** Yuhang Zhang, Baojiu Chen, Xizhen Zhang, Jinsu Zhang, Sai Xu, Xiangping Li, Yichao Wang, Yongze Cao, Lei Li, Hongquan Yu, Xin Wang, Duan Gao, Xuzhu Sha, Li Wang

**Affiliations:** School of Science, Dalian Maritime University, Dalian 116026, China; yuhangzhang@dlmu.edu.cn (Y.Z.); zhangxizhen@dlmu.edu.cn (X.Z.); melodyzjs@dlmu.edu.cn (J.Z.); xusai@dlmu.edu.cn (S.X.); lixp@dlmu.edu.cn (X.L.); wangyc2020@dlmu.edu.cn (Y.W.); cyz@dlmu.edu.cn (Y.C.); lilei@dlmu.edu.cn (L.L.); yuhq7808@dlmu.edu.cn (H.Y.); shadow7@dlmu.edu.cn (X.W.); liushengyi10@163.com (D.G.); hhhawn@icloud.com (X.S.); wangli201626@163.com (L.W.)

**Keywords:** optical gain, amplified spontaneous emission, Cu^+^, Tm^3+^, germanate glasses

## Abstract

Broadband tunable solid-state lasers continue to present challenges to scientists today. The gain medium is significant for realizing broadband tunable solid-state lasers. In this investigation, the optical gain performance for Tm^3+^ and Cu^+^ single-doped and co-doped germanate glasses with broadband emissions was studied via an amplified spontaneous emission (ASE) technique. It was found that the net optical gain coefficients (NOGCs) of Tm^3+^ single-doped glass were larger than those for Cu^+^ single-doped glass. When Tm^3+^ was introduced, the emission broadband width of Cu^+^-doped glass was effectively extended. Moreover, it was found that for the co-doped glass the NOGCs at the wavelengths for Tm^3+^ and Cu^+^ emissions were larger than those of Tm^3+^ and Cu^+^ single-doped glasses at the same wavelengths. In addition, the NOGC values of Tm^3+^ and Cu^+^ co-doped germanate glasses were of the same order of magnitude, and were maintained in a stable range at different wavelengths. These results indicate that the Tm^3+^ and Cu^+^ co-doped glasses studied may be a good candidate medium for broadband tunable solid-state lasers.

## 1. Introduction

In the 21st century, semiconductor lasers possessing high reliability, high brightness and excellent monochrome, have quickly led to an increasing number of applications in fiber communication, optical sensors, medicine, surgery, and so on [1,2,3]. With the rapid development in the field of optoelectronic information technology, the demand for information transmission systems with higher transmission capacity has also increased greatly [4,5]. To achieve a more efficient transmission network, an attractive method is to increase the number of transmission channels by broadening the gain bandwidth [6]. In addition, broadband tunable laser sources, which are widely used in scientific research, also require broadband solid-state gain media [7]. Therefore, there is growing interest in the development of continuous-wave (CW) broadband solid-state lasers.

As a key step in the development of CW broadband solid-state laser sources, the exploration of novel gain host materials with broadband and flat gain spectrum is indispensable. To this end, transition metal (TM)-ion-doped glass materials have attracted significant attention from researchers because the TM ions can easily enable broadband emission with full-width at half-maxima (FWHM) greater than 100 nm in a glassy host [8,9,10]. The copper ion, in particular, is characterized by a partially filled d shell and possesses outstanding fluorescence properties due to the presence of multiple valence and coordination states [11]. In our previous work, the fluorescence properties of Cu^+^- and Tm^3+^-doped germanate glasses were investigated, with a broadband emission ranging from 400 to 700 nm in the studied glasses observed, which could be useful for the implementation of white light sources [12]. The previously obtained results prompted us to consider that the Cu^+^- and Tm^3+^-doped germanate glasses may also possess broadband optical gain characteristics, with potential application prospects in visible CW broadband lasers. Therefore, the optical gain performance of Cu^+^- and Tm^3+^-doped germanate glasses was regarded as an important research goal to explore potential application in broadband lasers, which represented a different focus from the previous work.

In this study, a high-quality and acid-resistant germanate glass with lower phonon energy was investigated as a laser working medium. To examine the optical gain performance of Cu^+^- and Tm^3+^-doped germanate glasses, the net optical gain coefficients (NOGCs) for Cu^+^ and Tm^3+^ single-doped and co-doped samples were measured via a so-called amplified spontaneous emission (ASE) technique. Under 365 nm excitation, the relevant ASE spectra of Cu^+^ and Tm^3+^ single-doped and co-doped germanate glasses were measured. Furthermore, the NOGCs of Cu^+^ and Tm^3+^ single-doped and co-doped germanate glasses were calculated and compared. It was found that all the studied glasses presented positive optical gain coefficients.

## 2. Experiments

### 2.1. Materials and Synthesis

K_2_CO_3_ (Tianjin Kemiou Chemical Reagent Co., Ltd., Tianjin, China), Na_2_CO_3_ (Tianjin Kemiou Chemical Reagent Co., Ltd., Tianjin, China), ZnO (Tianjin Fuchen Chemical Reagent Factory, Tianjin, China), MgO (Tianjin Jinbei Fine Chemical Co., Ltd., Tianjin, China), Al_2_O_3_ (Tianjin Guangfu Fine Chemical Research Institute, Tianjin, China), GeO_2_ (Dalian Zhaoheng Technology Co., Ltd., Dalian, China), Cu(NO_3_)_2_·3H_2_O (Tianjin Fuchen Chemical Reagent Factory, Tianjin, China) and Tm_2_O_3_ (Sinopharm Chemical Reagent Co., Ltd., Shanghai, China) reagents were adopted as starting materials to synthesize Cu^+^- and Tm^3+^-doped germanate glass samples by a traditional melting-quenching method, with molar composition of 5K_2_O-12Na_2_O-3ZnO-5MgO-17Al_2_O_3_-58GeO_2_- *x*Cu(NO_3_)_2_-*y*Tm_2_O_3_. Here, *x* = 0.40 is for the Cu(NO_3_)_2_ single doping case (the obtained sample is denoted GC0.40); *y* = 0.25 is for Tm_2_O_3_ single doping case (the obtained sample is denoted GT0.25); *x* = 0.40, *y* = 0.25 is for Cu(NO_3_)_2_ and Tm_2_O_3_ co-doped samples (the obtained sample is denoted GC0.40T0.25). For the above samples, 4.00 wt% graphite powders (Tianjin Jinbei Fine Chemical Co., Ltd., Tianjin, China) of the designed glass weights were added as a reducing reagent to obtain univalent copper ions in air atmosphere. First, the raw materials were weighed based on the designed composition and mixed in an alumina crucible. Then, the well-mixed raw materials in the alumina crucible were put into an electric furnace (Tianjin Central Electric Furnace Co., Ltd., Tianjin, China) and heated at 1550 °C for 3 h. After that, the glass melt was poured into a preheated copper mold which was naturally cooled to room temperature. Finally, all the samples were annealed at 550 °C for 2 h, and the annealed glasses were cut and polished for the measurement of optical gain coefficients.

### 2.2. ASE Measurement

An optical gain effect is a prerequisite for materials to effect lasing action, and the NOGC is a necessary requirement for a laser working medium because it represents the existence of an optical signal amplified process which is the precondition for the laser operation [13]. For the acquisition of NOGC of the active materials, an ASE technique that is based on the relationship between ASE intensity and emission freely transmitting length (EFTL) can be adopted. As an optical process, ASE refers to the stimulated emission of an inverted atom triggered by a spontaneously emitted photon, which has been widely used in research concerning the laser medium in recent years [14,15]. Based on the above expression, a measuring system for ASE technique was established, as shown in Figure 1. A 365 nm light from a 150 W xenon lamp via a monochromator was used as a pump source, and the pump beam with uniform energy density was guided to a diaphragm placed vertically to the propagation direction of the 365 nm pump light. The slit height of the diaphragm was constant and the width was adjustable. The width could be read out from a micrometer. The glass sample was placed in parallel behind the diaphragm to accept the width-adjustable pump light. The ASE signal was detected on the glass end side direction which was orthogonal to the pump light propagation direction. By adjusting the pump beam width, the ASE spectrum for each corresponding pump beam width was recorded. Furthermore, the relation between the pump beam width and the relevant spectral intensity at certain wavelength could be established. The above measurements for the ASE spectra were carried out using a Hitachi F-4600 fluorescence spectrophotometer (Hitachi, Japan), and the built-in xenon lamp and two monochromators of the spectrophotometer were used without any location re-layout. The diaphragm and the sample to be measured were just set at the proper position in the sample chamber of the spectrophotometer.

## 3. Results and Discussion

### 3.1. Optical Gain of Cu^+^ Single-Doped Germanate Glass

As mentioned in the introduction, in a previous study [12], it was found that Cu^+^-doped germanate glass can generate a super-broadband emission which implies potential application for visible tunable lasers. In order to examine the feasibility of the broadband tunable laser operation of the studied glass, optical gain characterization of the Cu^+^-doped germanate glasses was performed using the setup shown in Figure 1. The EFTL of the 365 nm pump beam increased regularly from 0.1 to 0.7 mm with 0.1 mm steps. The ASE spectra of GC0.40 glass, under 365 nm excitation were obtained and plotted in Figure 2a. From Figure 2a, it can be seen that the ASE spectral intensity of GC0.40 glass gradually increased with EFTL increase. Meanwhile, each ASE spectrum of GC0.40 glass exhibited broadband emission, suggesting that the Cu^+^-doped germanate glass possessed potential for broadband tunable laser operation.

Under the above conditions, the NOGC at a certain wavelength λ is expressed as *g*, and the detected total emission intensity is denoted as *I*(Δ*l*) that involves the stimulated emission intensity Δ*I* and the spontaneous emission intensity *I*(Δ*l*)–Δ*I*, respectively. Therefore, using the relevance of EFTL Δ*l*, a definition equation for NOGC can be written as [14]
(1)ΔI=gI(Δl)Δl

Under the same conditions, the ASE intensity of the studied glass can also be presented as [16]
(2)$$\Delta I=I_{S}DT\Delta l\exp(g\Delta l)-I_{S}DT\Delta l$$
where *T* stands for thickness of the glass sample, *D* is the slit height of the pump beam which is formed by diaphragm hindering, and *I**_s_* is the excitation intensity per unit volume. From Equations (1) and (2), the relationship between total emission intensity *I*(Δ*l*) and NOGC *g* can be written as follows [17]
(3)$$I(\Delta l)=\frac{I_{S}DT}{g}\left[\exp(g\Delta l)-1\right]$$

In the above Equation (3), the total emission intensity *I*(Δ*l*) can be deduced from the ASE spectra, and the EFTL Δ*l* can be read from the diaphragm in Figure 1. Therefore, the dependence of *I*(Δ*l*) on Δ*l* at a specified wavelength λ can be extracted from the ASE spectra in Figure 2a. In this study, the dependences of *I*(Δ*l*) on Δ*l* for the wavelengths 500, 560 and 620 nm were derived and are depicted as the discrete data points in a single-logarithmic coordinates system as shown in Figure 2b. The experimental data (the discrete data points) were fitted to Equation (3), and the NOGCs at 500, 560 and 620 nm of GC0.40 glass were found in the fitting process to be 0.50, 0.74 and 1.05 cm^−1^, respectively. From these results, it can be concluded that optical gain can be realized in the Cu^+^-doped germanate glasses over a broad wavelength range, implying that the Cu^+^-doped germanate glass would be a good candidate for a broadband tunable laser working medium.

### 3.2. Optical Gain of Tm^3+^ Single-Doped Germanate Glass

Derived from the above analysis, it was found that the Cu^+^-doped germanate glass presented a net optical gain in wavelengths ranging from 450 nm to 700 nm. It was also found that there was still a lack of the blue component in the emission spectrum of the glass. Therefore, to broaden the wavelength range of the tunable laser, a luminescence center which can generate short wavelength emission should be introduced into the Cu^+^ single-doped germanate glass. It is well-known that the Tm^3+^ ion can emit blue light at around 450 and 480 nm corresponding to ^1^D_2_ → ^3^F_4_ and ^1^G_4_ → ^3^H_6_ transitions [18]. Therefore, in this study, the Tm^3+^ was taken to be the choice for supplying the blue emission in Cu^+^-doped germanate glass to extend the emission bandwidth. For the potential application of Tm^3+^-doped glass materials in lasers, most studies have focused on visible and infrared emissions based on an up-conversion technique [19,20,21], but the down-shifted visible emissions and correlated optical gain properties of Tm^3+^ are rarely reported.

To discover the optical gain performance of Tm^3+^ in germanate glass, the ASE spectra of GT0.25 glass were measured under 365 nm excitation and are shown in Figure 3a. From Figure 3a, two blue emission peaks, centered at 455 and 478 nm, were observed which can be assigned to ^1^D_2_ → ^3^F_4_ and ^1^G_4_ → ^3^H_6_ transitions of Tm^3+^, respectively. However, the 478 nm emission was much weaker than the 455 nm emission, implying that the non-radiative transition rate of the ^1^D_2_ level was relatively small since the population of the ^1^G_4_ level was achieved via non-radiative relaxation of the ^1^D_2_ level.

To quantitatively study the optical gain performance of the Tm^3+^-doped germanate glasses, the dependence of the ASE intensity on the EFTL for GT0.25 glass at 455 nm was derived from Figure 3a and is depicted in Figure 3b as discrete data points. By fitting Equation (3) to the experimental data, the corresponding NOGC was confirmed to be 4.67 cm^−1^, which was larger than that of the Cu^+^ single-doped glass, suggesting that the optical gain can also be effectively realized in the Tm^3+^ single-doped germanate glasses. In addition, the goodness of fit *R*^2^ was determined to be 0.999, which implies that the fitting result is reliable.

### 3.3. Optical Gain of Cu^+^ and Tm^3+^ Co-Doped Germanate Glass

In the above sections, it was observed that the net optical gains could be achieved from both the Tm^3+^ and Cu^+^ single-doped germanate glasses, but this does not mean that the same gain performance can be achieved in the Tm^3+^ and Cu^+^ co-doped glass. Therefore, it was necessary to further demonstrate the optical gain in the co-doped glass. To this end, the ASE spectra for the Tm^3+^ and Cu^+^ co-doped glass were measured under 365 nm excitation in an analogous way as described in above sections. Figure 4a shows the obtained ASE spectra of GC0.40T0.25 glass when the EFTL is changed from 0.1 to 0.7 mm with an increment of 0.1 mm.

From Figure 4a, it can be seen that each spectrum consisted of two parts: a broad band whose wavelength ranged from 500 to 700 nm, corresponding to the Cu^+^ emission, and a narrow strong band peaking at 455 nm and a weak one peaking at 478 nm, corresponding to the ^1^D_2_ → ^3^F_4_ and ^1^G_4_ → ^3^H_6_ transitions of Tm^3+^. In the spectral aspect, these two parts overlapped together, meaning that the emission bandwidth was satisfactorily extended in comparison with the Cu^+^ single-doped glass, as we expected. The FWHMs of Tm^3+^ and Cu^+^ in the co-doped glass were determined to be 15 and 135 nm, respectively. To determine the NOGCs for the Tm^3+^ and Cu^+^ co-doped glass, the dependences of the ASE spectrum intensities at 455, 500, 520, 540, 560 580, 600 and 620 nm on the EFTL were derived from the ASE spectra in Figure 4a and are shown in Figure 4b in the single-logarithmic coordinates system, respectively. Furthermore, Equation (3) was fitted to the data in Figure 4b in a similar way as described in Section 3.1 and 3.2. In the fitting processes, the NOGC at 455 nm was determined to be 5.37 cm^−1^, which was higher than that of the Tm^3+^ single-doped germanate glass, but the values for both single- and co-doped germanate glasses were of the same order of magnitude. From the fitting processes, the NOGCs at 500, 520, 540, 560 580, 600 and 620 nm were also determined to be 4.04, 5.36, 4.83, 4.10, 4.14, 4.16 and 5.34 cm^−1^, being much larger than that of the Cu^+^ single-doped glass, respectively. The differences between the NOGCs for the single-doped and co-doped glasses will be further explicated in future studies of the luminescence lifetime. Compared with the NOGCs at 2037 nm emission of the Ho^3+^/Tm^3+^/Ce^3+^ tri-doped tellurite glasses (1.13 cm^−1^) [22], at 570 nm emission of the Sm^3+^/silver-aggregate-doped borate glass (2.27 cm^−1^) [7], and at 1.8 μm emission of the Er^3+^/Tm^3+^ co-doped lead silicate glasses (1.12 cm^−1^) [23], the higher NOGCs of the Tm^3+^ and Cu^+^ co-doped samples indicate that the Tm^3+^ and Cu^+^ co-doped germanate glasses are a promising candidate for broadband tunable solid-state lasers. The relationship between NOGCs and wavelength for the co-doped glass is presented in Figure 4c. As can be seen, the NOGC values of Tm^3+^ and Cu^+^ co-doped germanate glasses stayed within a stable range at different wavelengths. The same order of magnitude of the optical gain coefficients for Tm^3+^ and Cu^+^ co-doped germanate glasses is beneficial to the practical application for laser operation since laser output can be realized when the laser device parameters are similar.

## 4. Conclusions

In summary, a series of Cu^+^- and Tm^3+^-doped germanate glasses were successfully prepared by a melt-quenching technique, and a direct measurement method was used to investigate the optical gain properties of the glass samples. Under 365 nm excitation, the NOGCs at 500, 560 and 620 nm for Cu^+^ single-doped germanate glass were found to be 0.50, 0.74 and 1.05 cm^−1^, respectively, and the NOGC at 455 nm for Tm^3+^ single-doped was calculated to be 4.67 cm^−1^. Under the same conditions, the NOGCs at 455, 500, 520, 540, 560 580, 600 and 620 nm for the Tm^3+^ and Cu^+^ co-doped glass were determined to be higher, being equal to 5.37, 4.04, 5.36, 4.83, 4.10, 4.14, 4.16 and 5.34 cm^−1^, respectively. Furthermore, the NOGC values of Tm^3+^ and Cu^+^ co-doped germanate glasses were maintained in a stable range at different wavelengths. The positive NOGCs imply that the studied glasses may be good laser working media for broadband tunable solid-state lasers.

## Figures and Tables

**Figure 1 materials-15-02134-f001:**
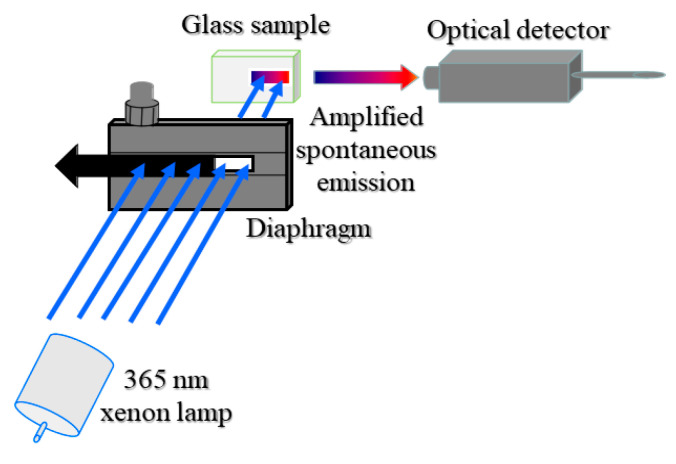
Setup of characterization system for optical gain.

**Figure 2 materials-15-02134-f002:**
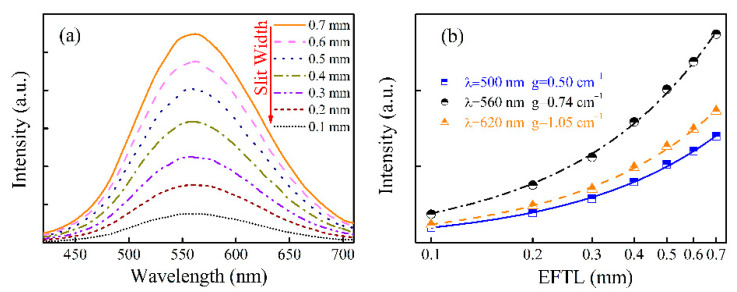
(**a**) ASE spectra of GC0.40 under 365 nm excitation when EFTL is changed from 0.1 to 0.7 mm. (**b**) Dependences of emission intensity on the wavelength and EFTL.

**Figure 3 materials-15-02134-f003:**
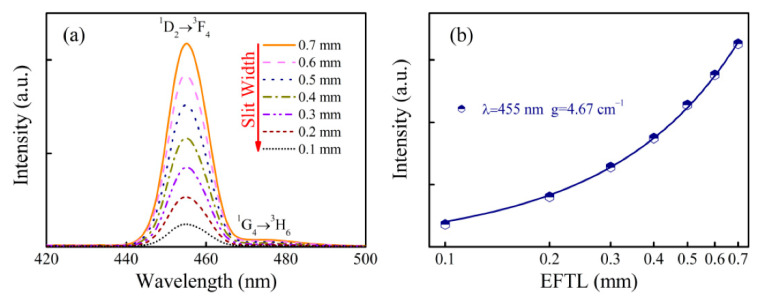
(**a**) ASE spectra of GT0.25 under 365 nm excitation when the EFTL is changed from 0.1 to 0.7 mm. (**b**) Dependences of emission intensity on the EFTL.

**Figure 4 materials-15-02134-f004:**
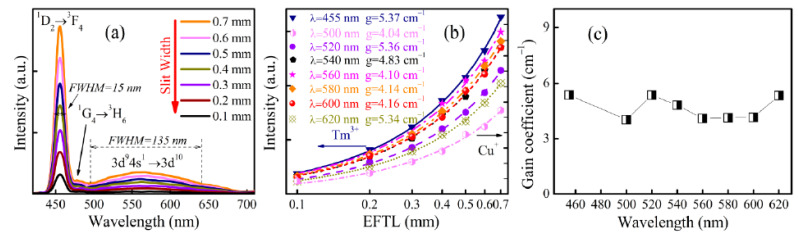
(**a**) Emission spectra of GC0.40T0.25 glass under the excitation of 365 nm when the EFTL is changed from 0.1 to 0.7 mm; (**b**) Dependences of emission intensity for Tm^3+^ and Cu^+^ on the wavelength and EFTL; (**c**) Relationship between NOGC and wavelength.

## Data Availability

The data presented in this study are available from the corresponding authors upon reasonable request.

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
