# Peer review of "Net Optical Gain Coefficients of Cu+ and Tm3+ Single-Doped and Co-Doped Germanate Glasses"

_materials, 2022, doi:10.3390/ma15062134_

Round 1

Reviewer 1 Report

The manuscript entitled "Net optical gain coefficients of Cu+ and Tm3+ single-doped and co-doped germanate glasses" presents a study on the optical properties of germanate glasses doped with Cu+ or Tm3+ as well as for the co-doped sample. The authors provide photoluminescence (PL) spectra of the studied samples and their detailed quantitative analysis. In conclusion, they state that the changes of PL from single-doped to co-doped glasses are probably owing to the changes of micro-environments surrounding the doping ions Tm3+ and Cu+.  I consider that such a statement needs to be confirmed by the PL decay curves for the single- and co-doped samples, so the manuscript should be updated with the PL decay data.

 PL (photoluminescence) decay curves are curves that graphically show luminescence lifetime, it is a common method in optical material science to obtain such data.   If authors are unavailable to perform this analysis then they should not speculate about the changes in the micro-environments and state that further understanding of the observed phenomena could be achieved during the future studies of luminescence lifetime.   

Author Response

R: Thanks for reviewer’s comments. Just as the reviewer has put forward, the changes of micro-environments surrounding the doping ions Tm3+ and Cu+ in the germanate glass need to be confirmed by the PL decay curves of single- and co-doped samples. Unfortunately, the test of photoluminescence decay curve can not be carried out at present due to the lack of characterization equipment. Based on the comments of reviewer, the relevant conclusions in the manuscript have been revised and stated that the observed change phenomena will be further understood during the future studies of luminescence lifetime. Please find the changes in the revised manuscript.

Reviewer 2 Report

  • In this work, a high-quality and acid-resistant germanate glass with lower phonon energy is investigated as a laser working medium.
  • The net optical gain coefficients for the Cu+ and Tm3+ single-doped and co-doped germanate glasses were measured by the amplified spontaneous emission (ASE) technique.
  • Under 365 nm excitation, the relevant ASE spectra of Cu+ and Tm3+ single-doped and co-doped germanate glasses were measured.

  • The net optical gain coefficients of Cu+ and Tm3+ single-doped and co-doped germanate glasses were calculated and compared.
  • It was found that all the studied glasses presented positive optical gain coefficients

In my opinion, the paper is suitable to be acceptable for publishing as it is now.

Author Response

R: Thanks for reviewer’s comments.

Reviewer 3 Report

The Author described studies concerning on net optical gain coefficients of copper and thulium single- and co-doped germinate glasses. The manuscript could be published in Materials after major revision.  Below, several aspects  have mentioned, which should be corrected and some doubts should be explained.

  1. The aim of the studies should be highlighted in the Introducion.
  2. All symbols in equations should be explained.
  3. The discussion is poor. The Authors should compare their results to data of others researchers.

Generally, the Authors did interest work. I recommend major revision.

Author Response

  1. The aim of the studies should be highlighted in the Introducion.

R: Thanks for reviewer’s comments. In our previous work, the fluorescence properties of Cu+ and Tm3+ doped germanate glasses were investigated, and meanwhile a broadband emission ranging from 400 to 700 nm in the studied glasses was observed, which could be useful for the implementation of white light sources. The previously results prompted us to realize that the Cu+ and Tm3+ doped germanate glasses may also possess broadband optical gain characteristic, which has potential application prospects in visible CW broadband lasers. Therefore, the optical gain performance of the Cu+ and Tm3+ doped germanate glasses was regarded as an important research goal to explore the potential application in broadband lasers filed, which is focused different from the previous work. To investigate the broadband optical gain performance of Cu+ and Tm3+ doped germanate glasses, the relevant studies were carried out.

According to the reviewer’s comments, the aim of the studies was further highlighted in the introduction. Please find the changes in the introduction.

  1. All symbols in equations should be explained.

R: Thanks for reviewer’s comments. All the symbols in the equations have been explained in the manuscript. Please find the changes in the revised manuscript.

  1. The discussion is poor. The Authors should compare their results to data of others researchers.

R: Based on the reviewer’s comments, a direct data comparison has been presented to better prove the conclusion, and the related discussions have been added in the revision. Compared with the net optical gain coefficients (NOGCs) at 2037 nm emission of the Ho3+/Tm3+/Ce3+ tri-doped tellurite glasses (1.13 cm-1) [22], at 570 nm emission of the Sm3+/silver aggregates doped borate glass (2.27 cm-1) [7], at 1.8 μm emission of the Er3+/Tm3+ co-doped lead silicate glasses (1.12 cm-1) [23], higher NOGCs of Tm3+ and Cu+ co-doped samples indicate that the Tm3+ and Cu+ co-doped germanate glasses are a promising candidate for broadband tunable solid state lasers. Please find the changes in the revised manuscript and Ref. [22], [7] and [23].

Refs:

[22] Yun C.; Zhang C.M.; Miao X.L.; Li Z.W.; Lai S.Y.; Sang T. Ultra-broadband 4.1 μm mid-infrared emission of Ho3+ realized by the introduction of Tm3+ and Ce3+. J. Lumin. 2021, 239, 118368.

[7] Zhong, H.; Chen, B.J.; Fu, S.B.; Li, X.P.; Zhang J.S.; Xu S.; Zhang Y.Q.; Tong L.L.; Sui G.Z.; Xia H.P. Broadband emission and flat optical gain glass containing Ag aggregates for tunable laser. J. Am. Ceram. Soc. 2019, 102, 1150-1156.

Reviewer 4 Report

This article discusses a study involving the optical gain performance of Tm3+ and Cu+ single-doped and co-doped germanate glasses with broadband emissions. It was found that net optical gain coefficients of the Tm3+ single-doped glass are larger than that of the Cu+ single-doped glass. This article appears suitable for the journal, although I have a few remarks.

1. Please provide references for the equations used in the manuscript.

2. Please do not use both red and green colors in the figures as some readers may not be able to tell the difference.

3. Please provide a good-of-fit for the fitted curve in Figure 3(b). 

Author Response

  1. Please provide references for the equations used in the manuscript.

R: Thanks for reviewer’s comments. The references related to the equations have been provided and added to the references. Please find the changes in the Ref. [14], [16] and [17].

Refs:

[14] Fu S.B.; Chen B.J.; Zhang J.S.; Li X.P.; Zhong H.; Tian B.N.; Wang Y.Z.; Sun M.; Zhang X.Q.; Cheng L.H.; Zhong H.Y.; Xia H.P. High upconversion optical gain of Er3+-doped tellurite glass. Appl. Phys. A 2014, 115, 1329-1333.

[16] Shaklee, K.L.; Nahaory, R.E.; Leheny, R.F. Optical gain in semiconductors. J. Lumin. 1973, 7, 284-309.

[17] Pavesi L, Negro LD, Mazzoleni C, Franzo G, Priolo F. Optical gain in silicon nanocrystals. Nature 2000, 408, 440-444.

  1. Please do not use both red and green colors in the figures as some readers may not be able to tell the difference.

R: Based on the reviewer’s comments, the red and green curves in the figure have been changed to other colors and curves have been modified to different line types to enable readers to clearly observe the differences. Please find the changes in the revised manuscript.

  1. Please provide a good-of-fit for the fitted curve in Figure 3 (b).

R: Thanks for reviewer’s comments. In the fitting process, the fit goodness R2 of the fitting curve in Figure 3 (b) was determined to be 0.99917, implying that the fitting result is reliable. And the relevant data and statements have been updated in the manuscript. Please find the changes in the revised manuscript.

Round 2

Reviewer 1 Report

The manuscript now can be accepted. 

Reviewer 3 Report

The Authors took into account all reviewers comments and improved the manuscript. The manuscript could be published in present version.